# Frozen sound: An ultra-low frequency and ultra-broadband non-reciprocal acoustic absorber

Anis Maddi [1] ✉, Come Olivier[1], Gaelle Poignand [1], Guillaume Penelet[1], Vincent Pagneux [1] & Yves Aurégan [1]

The absorption of airborne sound is still a subject of active research, and even more since the emergence of acoustic metamaterials. Although being sub-wavelength, the screen barriers developed so far cannot absorb more than 50% of an incident wave at very low frequencies (<100 Hz). Here, we explore the design of a subwavelength and broadband absorbing screen based on thermoacoustic energy conversion. The system consists of a porous layer kept at room temperature on one side while the other side is cooled down to a very low temperature using liquid nitrogen. At the absorbing screen, the sound wave experiences both a pressure jump caused by viscous drag, and a velocity jump caused by thermoacoustic energy conversion breaking reciprocity and allowing a one-sided absorption up to 95 % even in the infrasound regime. By overcoming the ordinary low frequency absorption limit, thermoacoustic effects open the door to the design of innovative devices.

The control and manipulation of waves is a field of research that receives considerable attention for its potential practical applications[1–3]. Among the most prominent topics, there is the absorption of waves which is regarded as one of the most challenging problems, particularly when the system is substantially shorter than the typical wavelength. This problem is further compelling for sound waves due to the inherent low absorption of materials. Typically, two types of problems are being addressed: the reflection and the transmission problem. In the former, the system is considered to be transmissionless and involves coupling the absorber with a reflective backing, such as a hard wall. On the latter, the system is characterized by waves that can be reflected and transmitted on both sides of a two-port. Although the objective is similar, the design of these absorbers is by no means alike. In a reflection problem, the target is to adjust the impedance of the absorber so that it matches the characteristic impedance of the waveguide, this can be achieved by using an absorber with an adjusted Q-factor, and in such a scenario, a perfect absorption can be achieved. Several systems have been theoretically and experimentally implemented and have demonstrated high performances, among the most studied solutions, passive and active resonant systems in various forms, such as Helmholtz resonators (HRs), decorated membrane resonators (DMRs), space-coil systems, and active loudspeakers[4–19]. To address the transmission problem, it is well-established that one-sided absorption cannot exceed 50% for mirror symmetric and transmissive absorbers. This constraint arises because most absorbers are only capable of producing either a pressure or velocity jump, while a combination of both is required for a perfect absorption, usually, they take the form of a HR or a resistive material, which respectively provide a velocity or pressure jump. To overcome this limitation, several innovative absorbers have been developed, which include (but not limited to) asymmetrical array of resonators, ventilated metamaterials, and DMRs[20–32]. Despite these remarkable achievements, most of the proposed devices are resonant dependent, and as a result, they only achieve a high level of absorption over a limited frequency range. Moreover, because of the size constraints, resonance-based systems can hardly achieve significant absorption at very low frequencies ($f < 100$ Hz). A possible solution to address the low frequency absorption would be to use resistive materials, in their usual form they provide only a pressure drop, and therefore are subjected to the aforementioned absorption limitation,

[1]Laboratoire d'Acoustique de l'Université du Mans (LAUM), UMR 6613, Institut d'Acoustique - Graduate School (IA-GS), CNRS, Le Mans Université, Le Mans, France. ✉e-mail: anis.maddi@univ-lemans.fr

however, if an additional physical process is added, such as it provides a velocity drop, a compact and broadband absorber can be designed.

As illustrated in Fig. 1a ultrathin resistive panels submitted to an acoustic wave are responsible for a pressure jump caused by viscous losses[33]. The resulting absorption of the acoustic energy is a broadband effect which remains efficient down to the zero-frequency limit. As stated above, such type of absorber can only absorb up to 50 % of the incident energy, although Coherent Perfect Absorption can be achieved by sending incident waves from both sides of the panel, if the amplitude and phase of incident waves are tuned adequately[33–37]. In principle, it is possible to increase the absorption of a single incident wave if a velocity jump is also experienced by gas parcels passing through the absorber. As illustrated in Fig. 1b, this can be achieved by imposing a steep temperature gradient along the resistive panel, which gives rise to thermoacoustic energy conversion caused by heat exchanges between the oscillating gas parcels and the solid frame of the material. As will be shown in the following a porous medium submitted to a temperature gradient along the axis of sound propagation can indeed be described as a discontinuity for both pressure and velocity, which will be used hereafter to maximize the absorption of a single incident wave. The thermoacoustic effect has been extensively studied for several decades as an alternative technology to the conventional heat engines and refrigerators[38,39]. It has not been thoroughly explored yet for the manipulation of sound waves, although recent works have shown that, when using a periodic arrangement of thermoacoustic cells, this effect can be used as the key element of an acoustic diode[40] or for the design of exotic wave scatterers[41–44]. More generally, the presence of heating/cooling in an acoustic waveguide/resonator generally leads to generation or dissipation of acoustic energy, which is a phenomenon essential in many applications, not only thermoacoustic engines but also, for example, combustion chambers in various technical devices, from rocket engines to household boilers[45–47].

The objective of this paper is to show that, if an acoustic wave impinges with normal incidence from one side of a compact resistive panel, then a strong cooling of its opposite side enables to overcome the intrinsic limit in absorption of 50 %. This gives rise to an efficient, broadband, sub-wavelength absorber that can even approach total absorption as far as the cold spot temperature is sufficiently low, such that the incident sound is "frozen" inside the sound barrier (the word "frozen" being used to convey both ideas of cooling and suppression of motion). It is shown from experiments that a broadband absorption of 85 % can be achieved with a single resistive sheet cooled at one side with liquid nitrogen, and that an absorption as high as 95 % can be achieved with two resistive sheets placed in series. This large absorption remains efficient for frequencies as low as 10 Hz in experiments, and actually even for lower frequencies given the non-resonant nature of the absorber.

## Results
### Theoretical description

The thermoacoustic effect refers here to a process of energy conversion caused by the interaction of a gas submitted to acoustic oscillations in a porous medium submitted to a temperature gradient. This process is satisfactorily described since the pioneering works of Rott[48] who developed a linear theory which is nowadays commonly employed for the design of so-called thermoacoustic engines[49]. A wide range of thermoacoustic engines has been investigated in the last decades[39], and such systems have demonstrated good performance[38,50,51] and potentiality for applications such as the recovery of waste heat. The key element of a thermoacoustic engine is a porous material, referred to as a stack. This porous material is surrounded by heat exchangers and placed inside an acoustic waveguide. A gas parcel submitted to acoustic oscillations experiences a thermodynamic cycle through the stack, which gives rise to amplification or damping of acoustic energy[52], depending on the sign of the temperature gradient imposed by the heat exchangers and on the phasing between pressure and velocity fluctuations.

In this study, attention is focused on the absorption and scattering properties of a thermoacoustic cell, which itself is made up of a stack

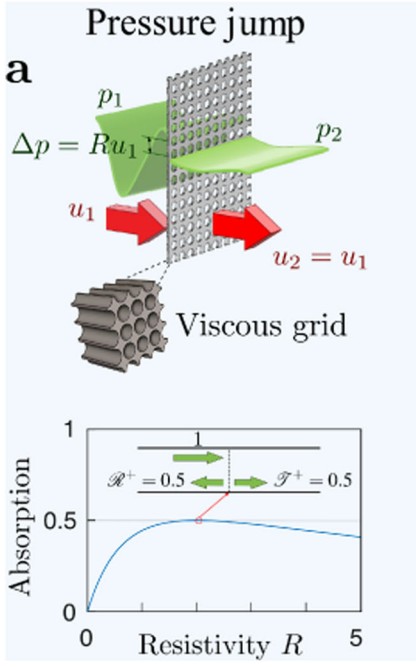

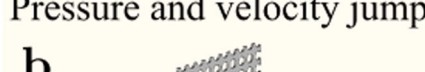

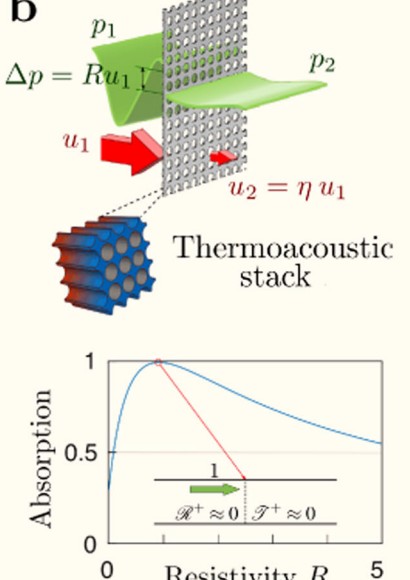

**Fig. 1 | Absorption of acoustic waves.** A comparison between a resistive material that delivers a viscosity-induced pressure jump, (**a**), versus the proposed concept based on the thermoacoustic effect, (**b**), providing both a pressure and velocity jump. In case (**a**), a maximum absorption of 50 % is achieved if the resistivity $R$ of the material equals 2. The corresponding reflexion and transmission coefficients, $\mathcal{R}^+$ and $\mathcal{T}^+$, are both equal to 0.5. In case (**b**), which is illustrated here for a velocity jump factor $\eta = 0.1$, an almost total absorption is achieved for a resistivity $R \approx 1$, and the system is non-reciprocal.

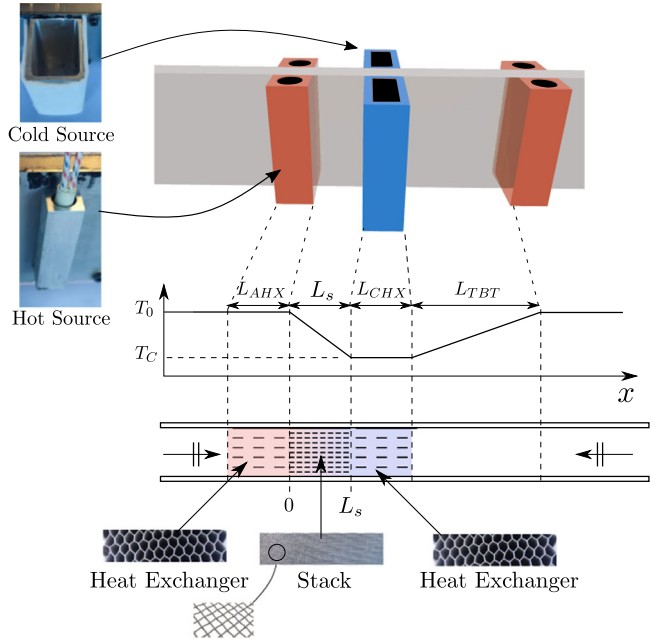

**Fig. 2 | Sketch of the thermoacoustic cell.** The cell includes the stack surrounded by hot and cold heat exchangers and the Thermal Buffer Tube. The stack consists of a pile of ultrathin stainless steel meshgrids, and the heat exchangers consist of a aluminum honeycomb material in thermal contact with the heat sources placed outside the waveguide. The hot source is an electric cartridge heater inserted in an aluminum block. The cold source is made up of a thermally insulated aluminum reservoir with liquid nitrogen inside. Further details on the complete experimental test-bench are provided in the Supplementary Information.

surrounded by heat exchangers and of the so-called Thermal Buffer Tube, as described in Fig. 2. According to the linear thermoacoustic theory[39,53], the stack is described as a medium made of parallel capillary tubes and the propagation of acoustic waves can be written as

$$\frac{dp}{dx} = -\underbrace{\frac{i\omega\rho_m}{S\phi(1-f_\nu)}}_{A}u,$$

$$\frac{du}{dx} = -\underbrace{\frac{i\omega S\phi}{\gamma p_m}\left[1+(\gamma-1)f_\kappa\right]}_{B}p + \underbrace{\frac{f_\kappa-f_\nu}{(1-f_\nu)(1-\sigma)}\frac{1}{T_m}\frac{dT_m}{dx}}_{C}u,$$

(1)

where $p$ is the acoustic pressure, $u$ is the acoustic volume velocity, and $\rho_m, T_m, p_m, \sigma$, and $\gamma$ stand for the mean fluid density, the mean temperature distribution, the mean pressure, the Prandtl number and the specific heat ratio of the fluid, respectively. The stack has a porosity $\phi$ and it is placed inside a waveguide of cross-sectional area $S$. The functions $f_\nu$ and $f_\kappa$ are complex valued, frequency-dependent functions[54], which describe the viscous and thermal coupling between the oscillating gas and the stack. For a fluid with a kinematic viscosity $\nu$ and a thermal diffusivity $\kappa$, we can define the viscous and thermal boundary layer thicknesses $\delta_\nu = \sqrt{2\nu/\omega}$ and $\delta_\kappa = \sqrt{2\kappa/\omega}$, such that for a porous material made of straight cylindrical pores with a radius $r$, the functions $f_\nu$ and $f_\kappa$ are given by:

$$f_{\nu,\kappa} = \frac{2}{(1-i)r/\delta_{\nu,\kappa}}\frac{J_1\left[(1-i)r/\delta_{\nu,\kappa}\right]}{J_0\left[(1-i)r/\delta_{\nu,\kappa}\right]},$$

(2)

where $J_0$ and $J_1$ are the zeroth and first order Bessel functions of the first kind. Eqs. (1) can be used to describe the acoustic propagation through the heat exchangers and the thermal buffer tube(TBT) by adapting the physical parameters to each section, since the only changes concern the porosity of each medium, the presence of a temperature gradient

or the size of the pores (just a single, wide pore for the TBT). As a result the transfer matrix of a complete thermoacoustic cell can be derived, as shown in the Supplementary Information.

Indeed, the constitutive equations, Eqs. (1), are actually very similar to the usual ones describing lossy propagation of acoustic waves through porous media[55,56]. The term $A$ in the first equation describes both inertia and viscosity effects, while the term $B$ in the second equation describes both compressibility effects and losses due to heat conduction. Moreover, for the specific case of a short material ($dx \rightarrow 0$) made of stacked meshgrids with very thin pores ($f_{\nu,\kappa} \rightarrow 1$) it is well-known that the material mostly provides a pressure drop $dp$ proportional to the volume velocity $u$ through a viscous resistance $A.dx$. However, as far as a temperature gradient is applied along the material, there exists an additional term, $C$, which describes thermoacoustic energy conversion and leads to a velocity jump $du$ proportional to the velocity $u$ through a factor $C.dx$. Therefore, attention should be focused on the stack which has a length $L_s$ and is (assumed to be) made up of many channels with radius $r_s$, the assumptions of a very short stack (i.e., $L_s \ll \lambda$ where $\lambda$ is the wavelength) and of a quasi-isothermal process (i.e., $r_s \ll \delta_{\nu,\kappa}$, such that $f_{\nu,\kappa} \approx 1 - i\frac{r_s^2}{4\delta_{\nu,\kappa}^2}$) enable to obtain a very simplified expression of its transfer matrix relating the normalized pressure $p^* = p/Z$ and the volume velocity $u$ at both sides of the stack (see the Supplementary Information for more details) :

$$\begin{pmatrix} p^* \\ u \end{pmatrix}_{L_s} = \underbrace{\begin{pmatrix} 1 & -R \\ 0 & \eta \end{pmatrix}}_{\mathbf{T}}\begin{pmatrix} p^* \\ u \end{pmatrix}_0,$$

(3)

where $Z = \rho_m c/S$ is the characteristic impedance of the waveguide ($c = \sqrt{\gamma p_m/\rho_m}$ is the speed of sound), $\eta = \frac{T_C}{T_0}$ the ratio of the right side temperature $T_C$ to the left side temperature $T_0$, and $R$ the resistivity of the material.

The scattering matrix $\mathbf{S}$ can also be derived from the transfer matrix $\mathbf{T}$ by decomposing the pressure and velocity into two traveling waves propagating in opposite directions, a right-going wave $p^+ = \frac{p+Zu}{2}$, and a left-going wave $p^- = \frac{p-Zu}{2}$, such that the matrix $\mathbf{S}$ satisfies

$$\begin{pmatrix} p_{L_s}^+ \\ p_0^- \end{pmatrix} = \underbrace{\begin{pmatrix} \mathcal{T}^+ & \mathcal{R}^- \\ \mathcal{R}^+ & \mathcal{T}^- \end{pmatrix}}_{\mathbf{S}}\begin{pmatrix} p_0^+ \\ p_{L_s}^- \end{pmatrix}.$$

(4)

The transmission and reflection coefficients are given by

$$\begin{aligned}
\mathcal{T}^+ &= \frac{2\eta}{1+\eta+R}, \\
\mathcal{T}^- &= \frac{2}{1+\eta+R}, \\
\mathcal{R}^- &= \frac{1-\eta+R}{1+\eta+R}, \\
\mathcal{R}^+ &= \frac{-1+\eta+R}{1+\eta+R}.
\end{aligned}$$

(5)

Therefore, in the presence of a temperature gradient i.e. $\eta \neq 1$, the system exhibits a nonreciprocal behavior[57] since $\mathcal{T}^+ \neq \mathcal{T}^-$.

Finally, we can introduce two absorption coefficients $\alpha^+$ and $\alpha^-$, such that

$$\alpha^+ = 1 - \frac{4\eta^2 + |\eta + R - 1|^2}{|1+\eta+R|^2},$$

(6)

$$\alpha^- = 1 - \frac{4 + |1 - \eta + R|^2}{|1+\eta+R|^2},$$

(7)

where $\alpha^+$ (resp. $\alpha^-$) is the absorption coefficient for a wave incident from the left (resp. from the right). In the following, since our main goal is to build an efficient, broadband and low frequency absorber,

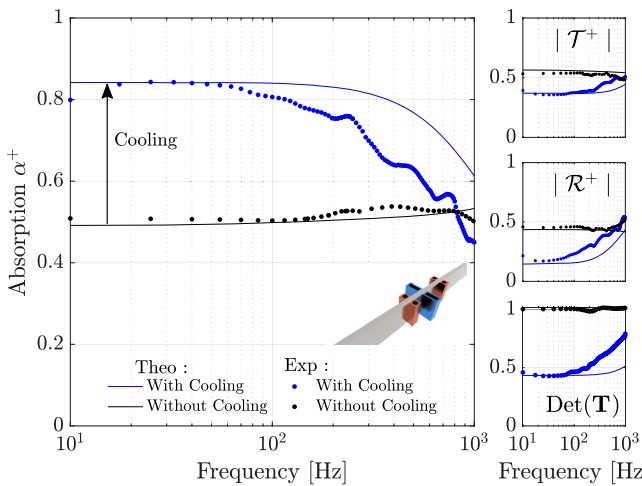

**Fig. 3 | Experimental validation for one cell.** The absorption coefficient, the transmission and reflection coefficients for a left-sided incident wave, and the determinant of the transfer matrix as functions of frequency. The solid blue lines correspond to the numerical model for $\eta = 0.42$ (which corresponds to the low frequency limit of det(**T**) obtained in experiments), while the blue markers (∘) correspond to the experimental data. The black lines and markers represent the theoretical and experimental results without cooling.

attention will be given to the absorption coefficient $\alpha^+$ when a left-incident wave experiences a temperature decrease through the stack. The quantities $\eta$ and $R$ are real-valued parameters, so that $\alpha^+$ is a continuous, real-valued, concave down function which reaches a maximum

$$\alpha^+_{max} = \frac{1}{\eta^2 + 1} \tag{8}$$

when $R = R_{opt} = 2\eta^2 - \eta + 1$. Hence, as long as the assumptions of a compact and quasi-isothermal ($r_s \ll \delta_\kappa$) stack hold, breaking the reciprocity by imposing a temperature difference ($\eta \neq 1$) and adequately tuning the resistivity $R$ of the stack enables to achieve a one-sided absorption higher than 50 %. For illustration, if the temperature is constant through the stack ($\eta = 1$), then the system is reciprocal and symmetrical, and the maximum absorption of $\alpha^+ = \alpha^- = 0.5$ is achieved for $R = 2$. However, if the system is cooled at the right side to a temperature $T_C$, supposedly that of liquid nitrogen such that a temperature ratio of $\eta \approx 0.25$ is achieved, then the system's reciprocity and symmetry are broken, giving rise to a much higher one-sided absorption of $\alpha^+ = 0.94$ (which is obtained for $R = 0.875$).

It is worth mentioning that a similar result can also be obtained by heating the left side of the stack rather than cooling its right side, but in order to reach the same absorption of about 0.94, a very high hot temperature of ≈1200 K (i.e. $\eta = 0.25$) is needed, which is not easy to achieve in practice.

The aforementioned absorption can be further enhanced while still using liquid nitrogen: a potential solution is to use two cells in series, such that the velocity jump provided by the two cells is equal to the product of the two (i.e., $\eta_{2cells} = \eta^2_{1cell}$). Hence, an absorption of $\alpha^+ \approx 0.996$ can be achieved theoretically.

## Experimental validation

A representation of a thermoacoustic cell with its associated temperature profile is given in Fig. 2. It consists of several heat exchangers, a stack with a decreasing temperature gradient (from the room temperature $T_0$ to the cold temperature $T_C$) and a passive duct portion called thermal buffer tube (TBT) along which the temperature increases (from the cold temperature $T_C$ to the room temperature $T_0$).

The cross-sectional area of the waveguide is $S = 5\,cm \times 1\,cm$. Heat can be supplied or removed from the system through the heat exchangers, using heat sources that are placed outside the waveguide, in thermal contact with the heat exchangers through the side walls (1 mm in thickness). The cold source consists of a pair of insulated aluminum reservoirs filled with liquid nitrogen. The hot source consists of cartridge heaters inserted in solid aluminum blocks which are placed at each end of the thermoacoustic cell to keep the temperature equal to the room temperature, see Fig. 2. The stack has a length $L_s = 16\,mm$, and it is made of stacked stainless steel wiremeshes, with an estimated porosity $\phi_s = 0.7$ and an estimated pore radius $r_s = 72\,\mu m$. Hereinafter, the experimental results obtained are shown for a system composed of either a single, or a series of two identical thermoacoustic cells. Further details regarding the measurements and the materials used are available in the Methods section, and in the Supplementary Information.

The measured variation of the absorption coefficient $\alpha^+$ is presented in Fig. 3 as a function of the frequency of the incident wave, for the case of a single TA cell. Black and blue markers correspond to the experimental results obtained at room temperature $T_0$, or when cooling of the right-side of the stack at temperature $T_C$ is applied, respectively. The corresponding transmission and reflection coefficients, $\mathcal{T}^+$ and $\mathcal{R}^+$, as well as the determinant of **T** are also presented in Fig. 3, and the low frequency limit det(**T**) ≈ $\eta$ of this determinant is used to determine the (indirectly controlled) temperature $T_C = \eta T_0$ of the right side of the stack. This knowledge of $\eta$ is used to compute the theoretical variations of $\alpha^+$, $\mathcal{T}^+$, $\mathcal{R}^+$ and det(**T**) as functions of the frequency, which correspond to solid black or blue lines (see Supplementary Information for further details on the model). In the absence of cooling with liquid nitrogen, the results show that approximately one half of the incident energy is absorbed by the meshgrids over the entire $10-1000$ Hz measurement band, corresponding likewise to transmission and reflection coefficients of ≈0.5, and since the system is reciprocal the determinant is around 1. It is worth noting that the experimental results are in good agreement with the numerical model, especially in the low frequency range. Once the heat sources are employed and a temperature drop is generated along the stack, the absorption coefficient increases from ≈0.5 to 0.84 at low frequencies, then decreases at higher frequencies as a result of the boundary layer becoming thinner, which affects the thermoacoustic energy conversion (i.e., the isothermal condition $r_s \ll \delta_{v,\kappa}$ is no longer satisfied). This additional absorption is also apparent in the reflection and transmission coefficients which decrease from ≈0.5 to 0.2 and 0.4, respectively. Likewise, the transfer matrix determinant diminishes to 0.42, indicating that thermoacoustic effects have indeed generated a velocity jump as well as non-reciprocal propagation. For the sake of clarity, the scattering coefficients of the incident waves from the opposite side are not shown but can be found in the Supplementary Fig. 6. If experimental results are compared to the numerical model, we can see that they follow the same behavior, but the model lacks accuracy at frequencies >100 Hz and tends to overestimate the absorption due to the thermoacoustic effect. This observed discrepancy for frequencies >100 Hz can be explained by several reasons, but the most plausible one is that the complex geometry of the stack (made with stacked meshgrids) is poorly described by the model (based on a parallel capillary tubes theory) and this inaccuracy in describing geometrical details is known to be more pronounced as far as the typical pore size approaches the viscous and thermal boundary layers thicknesses[39]. Moreover, the complexity of heat exchanges and therefore the actual temperature distribution in the system cannot be assessed very accurately . Nevertheless, the general trend predicted by the model globally conforms with the experimental results, especially for ultra-low frequencies.

As stated above, the low-frequency limit of det(**T**) provides an indication of the temperature ratio $\eta$ across the stack, which is found to be around $\eta \approx 0.42$, whereas a value $\eta \approx 0.25$ would have been

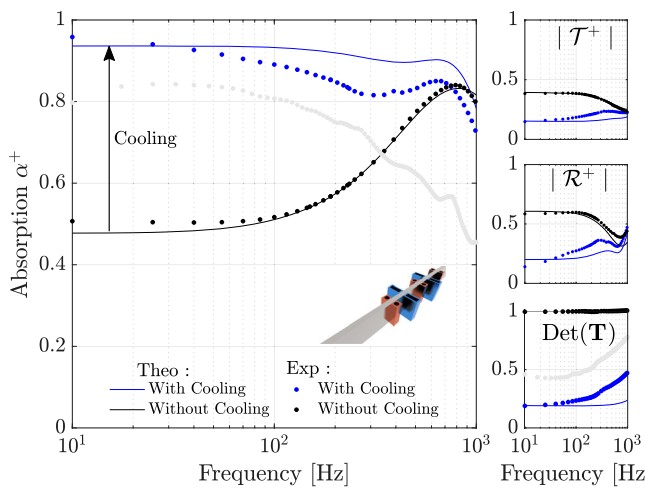

**Fig. 4 | Experimental validation for two cells.** The absorption coefficient, the transmission and reflection coefficients for a left-sided incident wave, and the determinant of the transfer matrix as functions of frequency. The solid blue lines correspond to the numerical model for $\eta = 0.42$, while the blue markers (∘) correspond to the experimental data. The black lines and markers represent the theoretical and experimental results without cooling. The soft gray markers correspond to a reminder of the experimental results presented in Fig. 3 for the case of a single cell with cooling.

expected in ideal conditions (based on the temperature of liquid nitrogen). Given that the system is compact, we decided to add a second cell, and the results for this configuration are shown in Fig. 4. If the absorber is kept at room temperature $T_0$, the absorption coefficient is roughly 0.5 at low frequencies but increases to 0.84 around 800 Hz owing to the Fabry-Perot interaction. We also notice a good agreement between the numerical and the experimental results. When heat sources are used, such that a temperature drop $\eta$ is applied to both cells, the absorption coefficient rises from 0.5 up to 0.94 for a frequency as low as 10 Hz, and remains >0.8 up to 800 Hz. This extra-absorption also translates into a reduction of the reflection and transmission coefficients reaching 0.15 for both. For this two-cells configuration, det(**T**) is as low as 0.22, which is approximately the squared value of the determinant in the one-cell configuration (i.e. $\det(\mathbf{T}_{2cell}) \approx \det(\mathbf{T}_{1cell})^2$), and therefore suggests that one might even add more cells to achieve a higher left-sided absorption. Much like the single cell, the numerical model does not accurately describe the system with a temperature drop at high frequencies, but it is still a useful tool to capture the overall trends.

## Zero frequency limit

The question of the (ultra) low-frequency limit below which the absorber may lose its efficiency is a natural and important question which could not be addressed here from experiments due to the limitations of our experimental set-up for frequencies <10 Hz (because of microphone spacing, as explained in the Supplementary Information). However, this limit can be analyzed using the linear thermoacoustic theory which can be used as a robust tool for describing thermoacoustic systems, even at frequencies <1 Hz[58]. Therefore, the low-frequency limit of the absorber was conducted numerically using the previous geometrical configuration for the one- and two-cells system (as for the results of the previous section, the numerical results are also based on finite-difference solving of Eqs. (1), as described in the Supplementary Information). The focus here is on frequencies below 10 Hz.

The predicted absorption is presented in Fig. 5 as a function of the frequency, where black or blue lines denote the system with $\eta = 1$ or $\eta = 0.42$, and where solid or dashed lines refer to the one- or the two-cell configuration, respectively. As anticipated, given the non-resonant

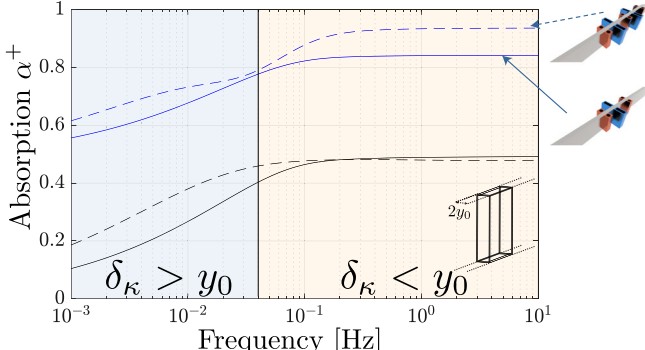

**Fig. 5 | Predicted absorption in the zero frequency limit.** A single (solid lines) or two thermoacoustic cells (dashed lines) at ultra-low frequencies. Black or blue lines refer to the system with $\eta = 1$ or $\eta = 0.42$, respectively.

nature of the absorber, the absorption coefficients remain fairly flat down to very small frequencies of about 0.1 Hz. Below such frequency the absorption decreases, and this decrease is caused by thermo-viscous effects occuring inside the Thermal Buffer Tube, as explained below. Should we introduce a typical length scale $y_0 = 5$ mm which is defined here as half the distance between the side-walls of the TBT (see. Fig. 5) it appears that a transition occurs when the frequency is such that the thermal boundary layer thickness $\delta_\kappa$ approaches $y_0$. If $\delta_\kappa \geq y_0$ the viscous losses and thermoacoustic effects play a key role in the TBT, which acts as a narrow channel for the incident waves. As a result, a left incident wave is strongly affected by both the stack and the TBT, which ultimately reduces the overall absorption. When heat sources are absent, the decrease in absorption also occurs at ultra-low frequencies, since the TBT is mostly a viscous resistance and the complete cell has a total resistance higher than its optimal value ($R = 2$). For the case of a large panel (absorbing panel) as the one depicted in Fig. 1(b), the low-frequency limit discussed above disappears (as $y_0$ becomes large). However, a low-frequency limit in the efficiency of absorption still exists, which involves the amplitude of the incident wave. The constitutive equations, Eqs. (1), only hold true for a gas parcel moving inside the stack, and they cannot properly describe a situation where the gas parcel displacement is larger than the length of the stack. For an incident pressure wave with a peak amplitude of 10 Pa (i.e., a sound pressure level of 114 dB SPL), the corresponding peak displacement at a frequency of 0.1 Hz is around 4 cm, which is larger than the stack length $L_s$. Still, the results show that the thermoacoustic absorber can achieve large absorption even for ultra-low frequencies, with corresponding wavelengths of up to a few kilometers: this might be useful for the absorption of infrasound[59].

It is worthwhile to note that the transmission loss of the thermoacoustic absorber is nearly constant in the infrasound range, and is given by $TL = -10 \log(|\mathcal{T}^+|^2) \approx 16.5$ dB (with $|\mathcal{T}^+| \approx 0.15$ according to Fig. 4). Let us compare to the transmission loss of a concrete wall with the same thickness as our device ($L = 15$ cm): the classical mass-law[60] states that the transmission loss tends to zero at ultralow frequencies, and increases to reach the value of 16.5 dB around 3 Hz. Hence, our system provides a better sound insulation than a concrete wall of the same thickness for $f < 3$ Hz. Additionally, in the infrasound range, the two-cell arrangement not only provides better sound insulation than a concrete wall, but also absorbs 95 % of the incident energy, as opposed to the wall which only reflects the acoustic waves.

## Discussion

One way of achieving a broadband acoustic absorption is to employ a short pile of stacked meshgrids which acts as a flow resistance that provides a pressure proportional to the volume velocity: for a transmission problem the maximum absorption cannot exceed 50 % and it

is achieved when $R = 2$. In this paper, we have shown that this maximum absorption can be exceeded if, in addition to the pressure jump caused by the viscous resistance, a volume velocity jump is generated through the absorber (i.e., by setting $\eta \neq 1$ in Eq. (3)). In practice, such a velocity jump can be achieved by applying a strong cooling on one side of the porous material. The resulting two-port becomes non-reciprocal, which allows the control of unidirectional waves, but above all it allows to get a higher absorption which is only limited by the temperature ratio imposed along the meshgrids. It is clear that the thermoacoustic effect offers the possibility to develop new kinds of absorbers capable of reaching high levels of absorption, even at extremely low frequencies. Currently, the only device able to deliver a similar pressure and velocity drop is an active system based on plasmacoustics, and has been recently used to design a broadband and highly effective absorber[61] in the reflection problem.

The experiments were performed by using liquid nitrogen as the cold heat source and the results show that a very large absorption of up to 95 % is achieved for a frequency as low as 10 Hz. Moreover, the broadbandness of the absorber, which is inherent to the compactness and non-resonating nature of the two-port, is confirmed by experiments since the absorption remains >80% over one decade for the one cell configuration, and >6 octaves for the two-cells configuration. The absorber has the advantages of being fairly compact and reasonably easy to implement, although it may also act as an amplifier for right incident waves (See Supplementary Fig. 6). Further improvements of the system could be achieved by optimizing the stack material and the heat exchangers to reach higher temperature ratios, and therefore higher absorption. Additionally, the present study focused exclusively on the absorption for a normal incidence, and could be extended to a more general problem, including oblique incidence. Another attractive aspect of the system studied here is its non-reciprocal behavior. Nowadays, systems with non-reciprocity are increasingly explored[57,62], as they offer a new facet for the design of innovative devices[63–70]. A weakness of the device considered here is that it is not easy to implement for real-life applications, as it requires imposing a steep temperature gradient along a porous material. However, the original properties of the thermoacoustic effect, namely those of both non-reciprocity and amplification/damping, could therefore be a promising way for the design of acoustic meta-materials.

## Methods

The absorber consists of either one cell or two cells in series, depending on the chosen configuration. Each cell is equipped with a stack of length $L_s = 16$ mm, made of stacked stainless steel wire meshes, with an estimated porosity $\phi_s = 0.7$ and an estimated pore radius $r_s = 74$ μm. A cold and an ambient heat exchangers are attached to each side of the stack: they both consist of a honeycombed aluminum material with a length $L = 1.5$ cm, made of several stacked layers of stainless steel wire meshes (Manufactured by Gantois, toile metallique 304LR n°167 FR0 056) a porosity $\phi = 0.945$, and a pore radius $r = 450$ μm. Heat can be supplied or removed from the system through those heat exchangers, using heat sources that are in thermal contact with the heat exchangers through the waveguide wall (1 mm in thickness). The cold source consists of a pair of insulated aluminum reservoirs of inner volume $V = 52$ cm³ filled with liquid nitrogen (see Fig. 2). During operation and due to the fast evaporation of liquid nitrogen, more liquid is regularly added to maintain the reservoirs full. The hot source consists of cartridge heaters inserted in solid aluminum blocks which are placed at the heat exchanger location on either side of the waveguide. Close to the hot source, a thermocouple was placed at the external face of the duct: it was used to monitor the hot temperature $T_0$ and to make so that this temperature stays close to the room temperature by adjusting the electrical power supplied to the cartridges. The last element of the thermoacoustic cell is the thermal buffer tube of length $L_{TBT} = 4.5$ cm along which the temperature

gradually increases up to room temperature. As a result, the total length of a single thermoacoustic cell is $L_{cell} = 7.1$ cm.

Measurements were performed up to a maximum frequency of 1 kHz, which is far below the cut-off frequency of the waveguide, estimated at $f_{co} = 3400$ Hz. Therefore, only plane waves are propagating along the duct. Furthermore, once the temperature distribution along the system has reached steady state (which takes a few minutes in the present system), a classical multiple microphones technique is used for the measurements of the system's transfer and scattering matrices. As displayed in the Supplementary Fig. 2, the complete test-bench includes two long ducts connected to the TA absorber, along which several microphones are mounted flush. Those ducts are also connected to an anechoic termination, and the system can be excited from either side using loudspeakers connected to each duct though a side-branch.

## Data availability

The measurement data that support the figures within this study are openly available on https://zenodo.org/record/7990401.

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

## Acknowledgements

This work was supported by the French National Agency for Research (SelfiXs Project, ANR-18-CE92-0001).

## Author contributions

A.M., C.O., G.Po. and G.Pe. designed the experiments. A.M. performed the experimental measurements and carried out the theoretical analysis. G.Po. and C.O. assisted with the experimental measurements. G.Pe., V.P. and Y.A. conceived and supervised the project. All authors contributed to the development of the concept, discussed the results and commented on the manuscript.

## Competing interests

The authors declare no competing interests.
