## [Peer Review File · Nature Communications]

Frozen sound: An ultra-low frequency and ultra-broadband non-reciprocal acoustic absorberReviewer #1 (Remarks to the Author):

A method for highly effective sound absorption at low frequencies and small spatial scale is proposed. The underlying theory is laid out and suitable experiments are conducted, whose results verify the claims.

To my knowledge, this is the first experimental realization of highly effective low-frequency sound absorption at sub-wavelength scale utilizing the thermoacoustic effect. The accompanying theory based on non-reciprocal acoustic wave propagation is introduced in a clear and concise manner.

Damping low-frequency acoustic waves is important for many technical applications, and intrinsically hard when space and weight are constraints (which is almost always the case). The present work demonstrates that it is possible to achieve high absorption rates at small scale when heating and/or cooling are added. The theory behind the experimental realization is well founded. The thermoacoustic effect is, of course, not new, but this particular application has not been explored before, as far as I am aware.

The experimental results nicely illustrate and support the theory. However, I do not find the 'low-frequency limit' part particularly useful. It is purely numerical (how exactly is it computed?), and there is no experimental evidence. I have some difficulties imagining somewhat effective absorption down to frequencies say below 0.1 Hz. I would remove this part until supporting experimental data is available. The low-frequency limit is, however, not essential to the main aspect of the manuscript.

Because of its novelty, clear theoretical foundation, convincing experimental verification and high relevance, I would support publication of this work. There are a few items that the authors may want to consider/correct, which I have listed below.

- Title: I am not sure what "Frozen sound" is supposed to refer to. The word 'frozen' does not appear anywhere in the main body, and the effect studied is not based on the effective speed of sound becoming very small.
- What could perhaps be added in the Introduction is something along the lines of the following: The fact that heating/cooling in an acoustic waveguide/resonator generally leads to generation or dissipation of acoustic energy is a phenomenon essential in many applications, not only thermoacoustic engines but also, for example, combustion chambers in various technical devices, from rocket engines to household boilers.
- The quantity η is defined as the ratio of the right side temperature to the left side temperature (II100-101). It is later mentioned (II121-122) that when the left side is heated to 1200 K, an η of 4 would be achieved. Based on the earlier definition of η this should be 1/4.
- In the discussion of Fig. 4, the absorption coefficient without cooling is said to reach 0.78 at frequencies around 800 Hz. However, it is clearly seen in the figure that it actually exceeds 0.8, maybe goes up to 0.83 at these frequencies.
- I am not entirely convinced that even with a 75 cm microphone spacing, the acoustic waves can be measured accurately down to 10 Hz, in particular with the chosen speakers that most likely do not have a great response at these low frequencies. Some more convincing arguments here would be appreciated.
- As explained above, I would suggest to remove the part on the low-frequency limit.

Reviewer #2 (Remarks to the Author):

In this paper, the authors have demonstrated a strategy to break the limitation that a subwavelength barrier cannot absorb more than 50% sound energy, via introducing a velocity jump caused by thermoacoustic energy conversion into the absorbing screen. Thus, a broadband

absorber capable of dissipating larger than 80% energy for 10-800 Hz sound is developed. The paper proposed a new methodology to develop sound absorber. However, I cannot recommend the paper to be published in Nature Communications otherwise the following issues are addressed.

1) The biggest concern of the paper is the conversion efficiency of the cold source such as liquid nitrogen and heat exchanger, which directly determines the feasibility and applicability of the designed sound absorber. Moreover, the real-time performance should be considered as it needs time for heat exchanger to achieve specific low or high temperature.

2) How the key parameter η is influenced by the temperature or even the concentration of liquid nitrogen inside the stack? In other words, how the absorption efficiency is influenced by the temperature gradient along propagation direction.

3) Why the absorption efficiency decreases in the high frequency region, such as near 1000 Hz in Fig. 3?

4) How is the absorption for sound wave under an oblique incident angle? This is an important issue especially for broadband metasurface given in Fig. 1 as in practical scenario, sound waves usually come from multiple directions but not just from normally-incident one.

Reviewer #3 (Remarks to the Author):

In this manuscript, the authors designed a subwavelength and broadband absorbing screen based on thermoacoustic energy conversion. The absorber consists of a porous layer with air at normal temperature at one side and cooled air at the other. The absorber can achieve a non-reciprocal absorption up to 95%, which addresses 50% absorption limitation at very low frequencies (< 100Hz) by other sound barriers. The novelty of the paper is doubted. The reviewer doesn't think the manuscript should be accepted by NC.

The main innovation of the paper lies in the introduction of thermoacoustic energy conversion effect into the sound absorber design. However, the thermoacoustic energy conversion is a well-known, extensively investigated physical phenomenon.

Besides, it is hard to apply the thermoacoustic energy conversion in real life passive sound absorber. The thermoacoustic effect is actually an active sound control method as it requires external energy to create temperature different at the two sides. What are the advantages of this active method over others?

Moreover, the sound absorbers are usually applied with rigid backings, such as on walls of tubes and on surfaces of panels. Can the thermoacoustic effect be applied in such situations?

The introduction should be improved, for example, the summary of sub-wavelength sound absorption is not comprehensive enough, there are much more structures that can absorb sound with substantially smaller dimensions than the wavelength except ultrathin resistive sheets and Helmholtz resonators.

Answers to Reviewers' comments

We thank the reviewers for their comments. In what follows, we repeat the comments in black, we respond in red, we cite from the original manuscript in black, and highlight changes made to the manuscript in blue.

Reviewer 1

A method for highly effective sound absorption at low frequencies and small spatial scale is proposed. The underlying theory is laid out and suitable experiments are conducted, whose results verify the claims.

To my knowledge, this is the first experimental realization of highly effective low-frequency sound absorption at sub-wavelength scale utilizing the thermoacoustic effect. The accompanying theory based on non-reciprocal acoustic wave propagation is introduced in a clear and concise manner.

Damping low-frequency acoustic waves is important for many technical applications, and intrinsically hard when space and weight are constraints (which is almost always the case). The present work demonstrates that it is possible to achieve high absorption rates at small scale when heating and/or cooling are added. The theory behind the experimental realization is well founded. The thermoacoustic effect is, of course, not new, but this particular application has not been explored before, as far as I am aware.

We are grateful to reviewer #1 for his/her overwhelmingly positive assessment of our manuscript.

The experimental results nicely illustrate and support the theory. However, I do not find the 'low-frequency limit' part particularly useful. It is purely numerical (how exactly is it computed?), and there is no experimental evidence. I have some difficulties imagining somewhat effective absorption down to frequencies say below 0.1 Hz. I would remove this part until supporting experimental data is available. The low-frequency limit is, however, not essential to the main aspect of the manuscript.

The low-frequency limit is the main concern of reviewer #1. The question of imagining how low can be the low-frequency limit below which the thermoacoustic effect can be used efficiently for the absorption of sound is actually an important question that arised naturally when we initiated those works. In what follows, we will address the referee's concern and try to convince him/her that although we could not make measurements down to such ultra-low frequencies, the linear thermoacoustic theory is robust and can be used instead to investigate the low-frequency limit.

Because of its novelty, clear theoretical foundation, convincing experimental verification and high relevance, I would support publication of this work. There are a few items that the authors may want to consider/correct, which I have listed below.

We are grateful to reviewer #1 for recognizing the novelty of our work.

Comment 1: Title: I am not sure what "Frozen sound" is supposed to refer to. The word 'frozen' does not appear anywhere in the main body, and the effect studied is not based on the effective speed of sound becoming very small.

Answer: The words "Frozen sound" in the title were not referring to some decrease in the speed of sound induced by a strong cooling of the propagation medium. In other words, we did not mean to make use of "slow sound" strategies to design the absorbing material (as used for instance in [J.P. Groby et al., J. Ac. Soc. Am. 139, 1660 (2016)]). We rather used those words to draw the picture of an incident sound wave which is "blocked" (or, say, "frozen") by the absorbing panel, and that this blockage is achieved by using liquid nitrogen (a "very cold" temperature). Hence, the term "frozen" somehow conveys both the idea of cooling and suppression of (acoustic) motion. This is the reason why we decided to use it in the title, and this is also why we would like to keep it.

Modification of the manuscript: to clarify what we meant by using the terms "frozen sound", we slightly modified the end of the introduction as follows (see modifications with blue fonts):

The objective of this paper is to show that, if an acoustic wave impinges with normal incidence from one side of a compact resistive panel, then a strong cooling of its opposite side enables to overcome the intrinsic limit in absorption of 50%. This gives rise to an efficient, broadband, sub-wavelength absorber that can even approach total absorption as far as the cold spot temperature is sufficiently low, such that the incident sound is "frozen" inside the sound barrier (the word "frozen" being used to convey both ideas of cooling and suppression of motion) . It is shown from experiments that a broadband absorption of 85% can be achieved with a single resistive sheet cooled at one side with liquid nitrogen, and that an absorption as high as 95% can be achieved with two resistive sheets placed in series.

Comment 2: What could perhaps be added in the Introduction is something along the lines of the following: The fact that heating/cooling in an acoustic waveguide/resonator generally leads to generation or dissipation of acoustic energy is a phenomenon essential in many applications, not only thermoacoustic engines but also, for example, combustion chambers in various technical devices, from rocket engines to household boilers.

Answer: Agreed. Thank you for this proposal.

Modification of the manuscript: the text is modified (see modifications with blue fonts) as follows:

... this effect can be used as the key element of an acoustic diode [21] or for the design of exotic wave scatterers [22–25]. More generally, the presence of heating/cooling in an acoustic waveguide/resonator generally leads to generation or dissipation of acoustic energy, which is a phenomenon essential in many applications, not only thermoacoustic engines but also, for example, combustion chambers in various technical devices, from rocket engines to household boilers[26-28].

and with added the references below:

26 :[Candel, S., Durox, D., Schuller, T., Bourgoïn, J. F., & Moeck, J. P. (2014). Dynamics of swirling flames. Annual review of fluid mechanics, 46, 147-173.]

27 : [Juniper, M. P., & Sujith, R. I. (2018). Sensitivity and nonlinearity of thermoacoustic oscillations. *Annual Review of Fluid Mechanics*, 50, 661-689.]

28 : [Lawn, C. J., & Penelet, G. (2018). Common features in the thermoacoustics of flames and engines. *International Journal of Spray and Combustion Dynamics*, 10(1), 3-37.]

Comment 3: - The quantity η is defined as the ratio of the right side temperature to the left side temperature (11100-101). It is later mentioned (11121-122) that when the left side is heated to 1200 K, an eta of 4 would be achieved. Based on the earlier definition of eta this should be 1/4.

Answer: It is indeed correct, the η would be equal to 1/4 if the the left side was heated to 1200K, we thank the referee for this remark.

Modification of the manuscript: the text is modified (see modifications with blue fonts) as follows:

...but in order to reach the same absorption of about 0.94, a very high hot temperature of approximately 1200K (i.e $\eta = 0.25$) is needed, which is not easy to achieve in practice.

Comment 4: In the discussion of Fig. 4, the absorption coefficient without cooling is said to reach 0.78 at frequencies around 800 Hz. However, it is clearly seen in the figure that it actually exceeds 0.8, maybe goes up to 0.83 at these frequencies.

Answer: Agreed, the referee is correct. The absorption coefficient around 800Hz reaches a maximum of 0.84.

Modification of the manuscript: the text is modified (see modifications with blue fonts) as follows:

...the absorption coefficient is roughly 0.5 at low frequencies but increases to 0.84 around 800Hz owing to the Fabry-Perot interaction.

Comment 5: I am not entirely convinced that even with a 75 cm microphone spacing, the acoustic waves can be measured accurately down to 10 Hz, in particular with the chosen speakers that most likely do not have a great response at these low frequencies. Some more convincing arguments here would be appreciated.

Answer: We do agree that a larger spacing would have been ideal for ensuring accurate measurements at low frequencies, but unfortunately this wasn't a possible solution with the test bench used here. However, we took maximum care to ensure a good accuracy of measurements down to 10 Hz. A careful relative calibration of the microphones was performed using a small cavity coupler. A step-by-step harmonic excitation was performed (Lock-in Amp), such that a good sound-to-noise ratio was achieved despite the potential limitation of the loudspeaker at low frequency. Additionally, this measurement bench was also used successfully for another study down to very low-frequency (see for instance, [Olivier,

Figure 1: **Coefficients of the transfer matrix of an empty duct obtained with a microphone spacing of 75 cm.** Points and straight line represents respectively the measured and analytical coefficients of the Transfer matrix of a duct (3 meters in length). The results are obtained here with a microphone spacing of 75 cm.

C. et al, Nonreciprocal and even Willis couplings in periodic thermoacoustic amplifiers. *Physical Review B*, 104(18), 184109.]). To make sure that we could make accurate measurements, we actually checked those measurements for the case of an empty waveguide of length 3 meters. The coefficients and the determinant of the transfer matrix are displayed in Fig.1, where we see a good agreement between the theoretical (straight lines) and experimental (dots) datas.

Modification of the manuscript: the text was modified in the supplementary information, in the paragraph that describes the experimental procedure (see modifications with blue fonts):

Note that a relative calibration of the microphones was performed beforehand using a small cavity coupler, so as to minimize measurement errors. The accuracy of the multiple microphones technique was also validated from the measurement of the transfer matrix of an empty duct (3 meters in length), and it was checked that the measured values of the T-matrix coefficients were matching the theoretical ones.

Comment 6: The experimental results nicely illustrate and support the theory. However, I do not find the 'low-frequency limit' part particularly useful. It is purely numerical (how exactly is it computed?), and there is no experimental evidence. I have some difficulties imagining somewhat effective absorption down to frequencies say below 0.1 Hz. I would remove this part until supporting experimental data is available. The low-frequency limit is, however, not essential to the main aspect of the manuscript.

- As explained above, I would suggest to remove the part on the low-frequency limit.

Answer: Actually, the question of the low-frequency limit below which the absorber might lose its efficiency arised naturally when we initiated those works and we think that it is worth being investigated.

This section was introduced to show that the absorber would still perform well until extremely low frequencies. It is true that we did not provide an experimental evidence of this, because of limitations of the test bench to measure at ultra-low frequencies, and we rather used the linear thermoacoustic theory. But as explained in the manuscript, this theory can be used as a reliable tool down to very low frequencies, as far as the peak-to-peak displacement does not exceed the length of the porous medium. Actually, this theory has already been used successfully to describe thermoacoustic engines with a very low working frequency (down to the infrasound range), like for instance those making use of liquid pistons: see for instance [Biwa, T., M. Prastowo, and E. Shoji. "Thermoacoustic modeling of Fluidyne engine with a gas-coupled water pumping line." *The J. Acoust. Soc. Am.* 152:2212-2219, (2022).] where such an engine working at a frequency of around 0.5 Hz is studied both theoretically and experimentally.

Therefore, we believe that this section should be kept in the main text. We also realized from the referee's comment that it was not specified in this section that the curves in Fig.5 were obtained using the finite-difference discretization of each component of a TA core, as done in the previous sections and as described with more details in supplementary material: this information is now clarified (and please note also that we obviously took care to choose a discretization step Δx to achieve the numerical convergence).

Modification of the manuscript: the beginning of the section devoted to the low frequency limit was modified as follows:

The question of the (ultra) low-frequency limit below which the absorber may lose its efficiency is a natural and important question which could not be addressed here from experiments due to the limitations of our experimental set-up for frequencies lower than 10Hz (because of microphone spacing, as explained in the Supplementary Information). However, this limit can be analyzed using the linear thermoacoustic theory which can be used as a robust tool for describing thermoacoustic systems, even at frequencies lower than 1 Hz [ref Biwa et al]. Therefore, the low-frequency limit of the absorber was conducted numerically using the previous geometrical configuration for the one- and two-cells system (as for the results of the previous section, the numerical results are also based on finite-difference solving of Eqs. (1), as described in the Supplementary Information). The focus here is on frequencies below 10Hz.

We also propose to provide a few more information in the supplementary information into the section called "Transfer matrix of a thermoacoustic cell" to clarify how exactly we computed this transfer matrix (see modification with blue fonts).

...If the temperature distribution is assigned at any position x along the thermoacoustic cell, and if each sub-component of a TA cell (stack, heat-exchangers, TBT) is itself discretized as a large number of elements of length Δx_n , then Eqs. (3)-(4) can be used to compute the transfer matrix M of each of the sub-components shown in Fig. 1. As a result,...

Reviewer 2

In this paper, the authors have demonstrated a strategy to break the limitation that a subwavelength barrier cannot absorb more than 50% sound energy, via introducing a velocity jump caused by thermoacoustic energy conversion into the absorbing screen. Thus, a broadband absorber capable of dissipating larger than 80% energy for 10 – 800 Hz sound is developed. The paper proposed a new methodology to develop sound absorber. However, I cannot recommend the paper to be published in Nature Communications otherwise the following issues are addressed.

We acknowledge reviewer #2 for pointing out that we have demonstrated a new strategy to break the limitation of 50% of a subwavelength barrier. In what follows we'll try to provide clear answers to the reviewer comments (and as far as possible we propose some modifications of the manuscript).

Comment 1 (part 1): The biggest concern of the paper is the conversion efficiency of the cold source such as liquid nitrogen and heat exchanger, which directly determines the feasibility and applicability of the designed sound absorber.

Answer: In this work, we have addressed the feasibility of the sound absorber, since we have built a system which enables to achieve the goal of a high absorption by a sound barrier at ultra-low frequency, thanks to a strong cooling at one side of the meshgrids. The question of the applicability of such a system for some real-life application is indeed another concern, which we did not try to address in this work. In other words, our goal was to provide the proof-of-concept of a *new* kind of absorbing material.

However we can give a few elements of answer regarding the conversion efficiency of the cold source, or more generally about the applicability of the system. First of all, we have designed a system which remains simple in terms of conception. The cold source is just a tank filled with liquid nitrogen, and the heat is pumped by this heat sink through the lateral walls (1mm in thickness) and the heat-exchanger which consists of aluminum honeycombed material in thermal contact with the meshgrids. Therefore, we just count on heat conduction through the heat exchanger and the lateral walls to pump heat from the inside to the heat sink. This is why the steady-state low temperature inside the system (at the right end of the meshgrids) is finally higher than the temperature of liquid nitrogen (otherwise we would have reached $\eta \approx 0.25$ rather than $\eta \approx 0.42$). A more efficient system could have been made if we had used a shell-and-tube heat exchanger made of many plates and crossed by tubes with flowing liquid nitrogen inside. Such a strategy could be employed by someone aiming at designing a large sound barrier based on the principle we have described here. The same remark also holds regarding the design of Ambient Heat Exchanger connected to a hot source which are used to keep the temperature at room temperature at the left side of the meshgrids. Finally, such considerations are some concerns about applied thermal engineering for some specific applications, which we did not consider in this work.

Comment 1 (part 2): Moreover, the real-time performance should be considered as it needs time for heat exchanger to achieve specific low or high temperature.

Answer: Indeed the heat exchanger need to achieve a specific temperature, but herein, the study was performed during the steady state that was reached after a few minutes. We agree that no mention of this was given in the paper, so we propose a correction in the Methods section. Here again, the consideration of the time needed to reach steady-state temperature

profile is rather a problem of heat transfer engineering which strongly depends on the system under consideration (note that in the end, for a carefully designed system where any heat leaks would be minimized through thermal insulation, both the cold temperature and the characteristic time for temperature stabilization could be lowered, and that the characteristic time would be mostly controlled by the thermal inertia of the meshgrids).

Modification of the manuscript (see modifications with blue fonts) in the Methods section :

Measurements were performed up to a maximum frequency of 1000 Hz, which is far below the cut-off frequency of the waveguide, estimated at $f_{co} = 3400$ Hz. Therefore, only plane waves are propagating along the duct. Furthermore, once the temperature distribution along the system has reached steady state (which takes a few minutes in the present system), a classical multiple microphones technique is used for the measurements of the system's transfer and scattering matrices...

Comment 2: How the key parameter η is influenced by the temperature or even the concentration of liquid nitrogen inside the stack? In other words, how the absorption efficiency is influenced by the temperature gradient along propagation direction.

Answer: As described in Eq.(3) in the frame of a *low frequency and thin pores approximation*, the parameter η is just equal to the ratio of the cold temperature to the room temperature T_C/T_0 . Therefore, in the low-frequency range a decrease in cold temperature will result in a decrease in η .

From our understanding regarding the comment about the concentration of liquid nitrogen inside the stack, there might be some misunderstanding: the liquid nitrogen is not introduced inside the stack, but it acts as a temperature sink that pumps heat through the lateral walls via an aluminum reservoir which is filled with liquid nitrogen (and kept filled during the whole experiments).

From our understanding the comment also refers to the impact of the *shape* of the temperature gradient along the duct axis. This can be investigated numerically using the finite-difference solving which is described in Supplementary Information, and which is also used to produce the numerical data in Figs.3-5. In the figure 2 below, we have calculated both α^+ and the coefficient T_{22} of the T-matrix of the TA cell (this coefficient is supposed to tend towards $\eta = T_C/T_0$ at low-frequencies) for three different shapes of the axial temperature distribution in the stack. The results show that for different temperature profiles and under the above stated conditions, the shape of the temperature distribution will not have a significant impact on the η parameter provided that the temperature difference remains the same (See Fig 2.a). However, the viscous resistance, as opposed to the parameter η , will be affected by the temperature profile (since the viscosity of the fluid depends on temperature) and this may improve (or decrease) the absorption depending on how close (or far) this value gets in comparison to the optimal viscous resistance (as shown in Fig 2.b).

So, to summarize, the axial profile of the temperature distribution does not impact the value of T_{22} which equals η in the low frequency range, but it slightly impacts the viscous resistance of the meshgrids, and therefore it slightly impacts the absorption. However, it is seen here that this effect of the temperature profile is weak. In our experiments we could not measure the temperature along the thermoacoustic core, but regarding the shortness of the core with respect to the

(a) Coefficient T_{22} of the transfer matrix for three temperature profiles. This coefficient is equal to the parameter η at low frequencies.

(b) Absorption coefficient α as function of the frequency, for 3 different temperature profiles.

Figure 2: Impact of the temperature gradient on the parameter η and the absorption coefficient α .

wavelength (at low frequencies which are the main focus of our works) this is not a problem, and the assumption of a linear temperature distribution is reasonable.

Modification of the manuscript : Based on this comment from reviewer # 2, we decided to include an additional section in the Supplementary Information which includes the figure discussed above and therefore gives more details regarding the impact of the shape of the temperature gradient on the efficiency of the absorber. This new section is reproduced below.

I.C. On the impact of the shape of the temperature distribution

As the shape of the temperature distribution within the TA cell cannot be measured, we checked numerically that it does not strongly impact the operation of the system in the low frequency range. The results are presented in Fig.2, where we have calculated both α^+ and the coefficient T_{22} of the T-matrix of the TA cell (this coefficient is supposed to tend towards $\eta = T_C/T_0$ at low-frequencies) for three different shapes of the axial temperature distribution in the stack. The results show that for different temperature profiles and under the above stated conditions, the shape of the temperature distribution will not have a significant impact on the η parameter provided that the temperature difference remains the same (See Fig 2.a). However, the viscous resistance, as opposed to the parameter η , will be affected by the temperature profile (since the viscosity of the fluid depends on temperature) and this may improve (or decrease) the absorption depending on how close (or far) this value gets in comparison to the optimal viscous resistance (as shown in Fig 2.b).

Comment 3: Why the absorption efficiency decreases in the high frequency region, such as near 1000 Hz in Fig. 3?

Answer: Higher frequency implies smaller boundary layer thickness $\delta_{k,v}$, and since the TA absorber requires to have pore radius way smaller than the boundary layer $r_p \ll \delta_{k,v}$, this conditions will eventually no longer be satisfied. Or in other words, increasing the frequency while keeping the pore radius constant leads to a weaker thermoacoustic energy conversion.

Modification of the manuscript (line 151-153)

...the absorption coefficient increases from approximately 0.5 to 0.84 at low frequencies, then decreases at higher frequencies as a result of the boundary layer becoming thinner, which affects the thermoacoustic energy conversion (i.e., the isothermal condition $r_s \ll \delta_{\nu,\kappa}$ is no longer satisfied). This additional absorption is also apparent in the reflection...

Comment 4: How is the absorption for sound wave under an oblique incident angle? This is an important issue especially for broadband metasurface given in Fig. 1 as in practical scenario, sound waves usually come from multiple directions but not just from normally-incident one.

Answer:

This study focused exclusively on the absorption for a normal incidence. However, in the case of an oblique incidence, some considerations can be drawn beforehand:

- For an incidence with an angle θ , the characteristic impedance will change from Z to $Z/\cos(\theta)$.
- Thus, for small angles, the impedance will not be much affected, and therefore the absorption will not be much affected either.

Therefore, the absorption should remain high for small angles, and a more in-depth study is needed to assess the performance of the absorber, especially for the case of a grazing incidence. Actually, this can be the subject of a study by itself and is beyond the scope of this present work.

Modification of the manuscript : We decided add a few sentences in the discussion sections about the absorption for an oblique incidence.

...Further improvements of the system could be achieved by optimizing the stack material and the heat exchangers to reach higher temperature ratios, and therefore higher absorption. **Additionally, the present study focused exclusively on the absorption for a normal incidence, and could be extended to a more general problem, including oblique incidence.** Another attractive aspect of the system studied ...

Reviewer 3

In this manuscript, the authors designed a subwavelength and broadband absorbing screen based on thermoacoustic energy conversion. The absorber consists of a porous layer with air at normal temperature at one side and cooled air at the other. The absorber can achieve a non-reciprocal absorption up to 95%, which addresses 50% absorption limitation at very low frequencies (< 100Hz) by other sound barriers. The novelty of the paper is doubted. The reviewer doesn't think the manuscript should be accepted by NC.

Comment 1:The main innovation of the paper lies in the introduction of thermoacoustic energy conversion effect into the sound absorber design. However, the thermoacoustic energy conversion is a well-known, extensively investigated physical phenomenon.

Answer: We must confess that we are puzzled by this strange comment. The thermoacoustic energy conversion is indeed extensively studied since several decades (notably by some of the co-authors), but to the best of our knowledge it has not yet been used for the absorption of sound. Moreover, this thermoacoustic effect has the unique feature (as far as we know) that it can provide a velocity jump proportional to the input velocity. This latter effect can be used advantageously

and the novelty of the works described in this paper lies in the fact that we propose a *new* solution that enables to overcome the 50% absorption limitation at very low frequencies for a wave *incident from one side*.

Therefore, we strongly disagree with the opinion of the referee that our work lacks of novelty. We can hardly accept an argument stating that, because the thermoacoustic effect is well-known since several years then, whatever you do with it, will lead to something well-known...

Comment 2: Besides, it is hard to apply the thermoacoustic energy conversion in real life passive sound absorber. The thermoacoustic effect is actually an active sound control method as it requires external energy to create temperature different at the two sides. What are the advantages of this active method over others?

Answer: The device considered here can be defined as active as would be any other system needing some external input of energy like, for instance, a one making use of a mean flow. Now, we hope that the referee will agree that it is different from a system based on active *control* where a signal from the incident wave would be captured by a sensor and then fed back (with adequate filtering) to a source.

In principle, a similar system based on active control with a *velocity* sensor placed in front of a monopolar point source, with the sensor feeding back its signal to the source, would be similar to the device we are considering, which provides a velocity jump proportional to the input velocity (in addition to the pressure jump proportional to the velocity). Such a method based on active control could be as well implemented in real-life situation, as far as

- one has an acoustic velocity sensor (pressure sensors are much more common),
- some solution to prevent from the onset of self-oscillations due to the audio feedback is addressed (which might not be easy because of time lags in the feedback loop),
- and, in the very low frequency range, the source can achieve large displacements (standard moving coil loudspeakers have a limited excursion of only a few millimeters), and it has a flat response with respect to frequency (which is usually not the case such that it should be corrected in the feedback loop)

. As we did not find in the literature some description of such a similar device based on a source-sensor-feedback system targeting to absorb the energy of a wave incident from one side (in a transmissive configuration) and at ultra-low frequency, we can hardly provide a formal comparison with our system. It seems that in both cases, the application for a real-life situation is not easy, but it is feasible (at least we have proven that our system is feasible).

Now, if the question is about the advantages of our system with regards to *completely passive* systems, we do not dispute that the latter passive systems, in general, should probably be more easy to implement for real-life applications. But on the other hand, if we consider a transmission problem with a wave incident from one-side, we have explained in the introduction that the latter passive systems cannot achieve more than 50 % absorption in the sub-wavelength regime unless their symmetry or reciprocity is broken... We have explained that in our system we have broken the reciprocity, and as far as we know, breaking the reciprocity will necessarily lead to some increase in the complexity of the system (whether one makes use of nonlinear effects, active control with feedback loop, spatio-temporal modulation or any action requiring external supply of energy).

Modification of the manuscript: Trying to address the criticism of the referee regarding the ease of implementation of wave scatterers based on the thermoacoustic effect (and also to address the similar comment of referee 2 regarding the applicability of the system), we propose to modify the end of the conclusion as follows:

A weakness of the device considered here is that it is not easy to implement for real-life applications, as it requires imposing a steep temperature gradient along a porous material. However, the original properties of the thermoacoustic effect, namely those of both non-reciprocity and amplification/damping, can be used as an additional mean for the design of acoustic meta-materials.

Comment 3: Moreover, the sound absorbers are usually applied with rigid backings, such as on walls of tubes and on surfaces of panels. Can the thermoacoustic effect be applied in such situations?

Answer: The problem we are considering is not the one of an absorbing material that should be placed in front of a rigid wall. It is rather another problem where we try to make a sound barrier, *therefore a two-port*, which would neither transmit nor reflect the sound impinging *from one side*. It is only for that kind of problem that there exists an intrinsic limit of 50% in absorption for a symmetric and reciprocal point scatterer (since coherent perfect absorption can be achieved even at ultra-low frequency if incident waves arrive from both sides). We also would like to point out that the transmission problem we are considering has already been considered by other authors, e.g. in [Yang-et-al-2015, merkel-et-al-2015, long-et-al-2021, jiminez-et-al-2017, Lee-et-al-2019, Tsuruta-et-al-2022], with the same goal of vanishing both the transmission and the reflection coefficients.

The referee asks whether the thermoacoustic effect can be applied for sub-wavelength absorption in such situations where the absorber is used with rigid backings, and the answer is no (unless, perhaps, the design of the system is thought differently to account for the presence of a rigid backing...). The presence of a rigid backing would indeed lead to a decrease of the absorption coefficient in the ultra-low frequency range, as it does for the simpler case of ultrathin meshgrids, which cannot act as a viscous resistance if they are placed close to a rigid wall.

[Yang-et-al-2015] Min Yang, Chong Meng, Caixing Fu, Yong Li, Zhiyu Yang, and Ping Sheng, Subwavelength total acoustic absorption with degenerate resonators, *Appl. Phys. Lett.* 107, 104104 (2015).

[merkel-et-al-2015] A. Merkel, G. Theocharis, O. Richoux, V. Romero-Garcia, and V. Pagneux, Control of acoustic absorption in one-dimensional scattering by resonant scatterers, *Appl. Phys. Lett.* 107, 244102 (2015).

[long-et-al-2021] Houyou Long, Chen Shao, Ying Cheng, Jiancheng Tao, and Xiaojun Liu, High absorption asymmetry enabled by a deep-subwavelength ventilated sound absorber, *Appl. Phys. Lett.* 118, 263502 (2021).

[jiminez-et-al-2017] N. Jiménez, et al., Rainbow-trapping absorbers: broadband, perfect and asymmetric sound absorption by subwavelength panels for transmission problems, *Sci.Rep.* 7 (1) (2017) 13595.

[Lee-et-al-2019] Lee, T., Nomura, T. & Iizuka, H. Damped resonance for broadband acoustic absorption in one-port and two-port systems. *Sci. Rep.* 9, 1–11 (2019).

[Tsuruta-et-al-2022] Tsuruta, R., Li, X., Yu, Z., Iizuka, H., & Lee, T. (2022). Reconfigurable Acoustic Absorber Comprising Flexible Tubular Resonators for Broadband Sound Absorption. *Physical Review Applied*, 18(1), 014055.

In order to answer more precisely to the referee's question, we have plotted in Fig.3 below the absorption coefficient α^+ as a function of the frequency for a one-cell configuration, and for different cases, namely:

- with filled-black markers, a one-cell configuration without cooling (same data as those of Fig.3 in the manuscript)
- with empty-black markers, a one-cell configuration without cooling *and with a rigid wall placed just behind the TA-cell* (the absorption coefficient can be obtained from the impedance just in front of the absorber, which itself is expressed as a function of the T-matrix coefficients of the TA-cell)
- with filled-blue markers, a one-cell configuration with cooling (same data as those of Fig.3 in the manuscript)
- with empty-blue markers, a one-cell configuration with cooling *and with a rigid wall placed just behind the TA-cell*

The results show that

- 1.- The presence of a rigid backing strongly impacts the absorption,
- 2.- The flat and broadband absorption at ultra-low frequency disappears if a rigid backing is placed behind the TA-cell, no matter if cooling is applied or not.
- 3.- if a rigid backing is placed behind the TA-cell, there exists a frequency range where a very high absorption can be achieved (especially if cooling is applied).

Modification of the manuscript: Based on this comment from reviewer #3, we decided to include an additional section in the Supplementary Information which includes the figure discussed above, where attention is focused on the effect of a rigid backing. This new section is reproduced below.

II.D. Effect of rigid backing

Although it falls out of the scope of this study where a transmission problem is considered, a question which may arise regarding the efficiency of the proposed non-reciprocal absorber is that of the impact a rigid backing which would be placed just behind the absorber. Measurements were not performed for such a configuration, but they can be used to predict the absorption of an incident wave by a TA cell and a rigid backing which imposes a vanishing velocity at the output of the absorber. The results obtained are presented in Fig. 3 for the case of a single TA cell. The absorption due to the TA cell equipped with a rigid backing are presented with blue or black open circle markers (depending on if cooling is applied or not) while the results for a transmission problem are provided with filled circle markers as a reminder. The main conclusion that can be drawn is that the broadband absorption at ultra-low frequency disappears if a rigid backing is placed behind the TA-cell, no matter if cooling is applied or not. Such a decrease of the absorption coefficient in the low frequency range is expected, as it is for the simpler case of ultrathin meshgrids which cannot act as a viscous resistance if they are placed too close to a rigid wall. It is worth noting that if a rigid backing is placed behind the TA-cell, there exists a frequency range where a very high absorption can be achieved (especially if cooling is applied).

Figure 3: Absorption coefficient α^+ for a one-cell configuration as a function of the frequency, depending on if a rigid wall is placed just behind the TA-cell (where $\alpha^+ = 1 - |\mathcal{R}^+ + \mathcal{T}^+ \mathcal{T}^- (1 + \sum_{n=1}^{\infty} (R^-)^n)|^2$ for a reflection problem) or if there is no rigid backing (transmission problem, where $\alpha^+ = 1 - |\mathcal{T}^-|^2 - |\mathcal{R}^+|^2$). Blue markers refer to the results obtained with cooling, while black markers refer to the results obtained without cooling, and the results are obtained from the measurements of the scattering matrix for one-cell.

Comment 4: The introduction should be improved, for example, the summary of sub-wavelength sound absorption is not comprehensive enough, there are much more structures that can absorb sound with substantially smaller dimensions than the wavelength except ultrathin resistive sheets and Helmholtz resonators.

Answer: In our introduction we wrote that *Possible solutions consist in using ultrathin resistive sheets or using panels composed of resonant systems, e.g. Helmholtz resonators with moderate Q-factor or membranes.* Therefore we did not only consider Helmholtz resonators but more generally resonant systems, which indeed can take a large number of forms and names (membranes, DMR, space-coil system...). We agree with the referee that it could be clarified, and that additional references are worth to be added in the manuscript. Therefore we changed the introduction and added some references, including those based on active control strategies.

Modification of the manuscript: The introduction (paragraph 1) was modified as follows

The control and manipulation of waves is a field of research that receives considerable attention for its potential practical applications[1–3]. Among the most prominent topics, there is the absorption of waves which is regarded as one of the most challenging problems, particularly when the system is substantially shorter than the typical wavelength.

Typically, two types of problems are being addressed: the pure reflection and the reflection + transmission problem. For the reflection problem, the absorber is backed by a rigid wall and only the minimisation of the reflected energy is studied. Then, the target is to adjust the impedance of the absorber so that it matches the characteristic impedance of the waveguide. When this occurs, perfect absorption can be achieved. Several systems have been theoretically and experimentally implemented and have demonstrated high performances, among the most studied solutions, passive and active resonant systems in various forms, such as Helmholtz resonators (HRs), decorated membrane resonators (DMRs), space-coil systems, and active loudspeakers [4–18]. In the reflection and transmission problem, it is established that, when sending the wave to one side only, the absorption cannot exceed 50% for an absorber much smaller than the wavelength. [19] This limit results from the fact that classical absorbers are only capable of producing a pressure jump (e.g. resistive material) or velocity jump (e.g. HRs), whereas a combination of the two is required to achieve perfect absorption. To overcome this limitation, several innovative absorbers have been developed, including (but not limited to) asymmetric resonator arrays, ventilated metamaterials and DMRs [20–32]. Despite these remarkable achievements, most of the proposed devices are resonant dependent, and as a result, they only achieve a high level of absorption over a limited frequency range. Moreover, because of the size constraints, resonance-based systems can hardly achieve significant absorption at very low frequencies ($f < 100$ Hz). A possible solution to address the low frequency absorption would be to use resistive materials. In their usual form they provide only a pressure drop, and therefore are subjected to the aforementioned absorption limitation. However, if an additional physical process is added, such as it provides a velocity drop, a compact and broadband absorber can be designed.

References

- [1] Cummer, S. A., Christensen, J. & Alù, A. Controlling sound with acoustic metamaterials. *Nat. Rev. Mater.* **1**, 1–13 (2016).
- [2] Ma, G. & Sheng, P. Acoustic metamaterials: From local resonances to broad horizons. *Sci. Adv.* **2**, e1501595 (2016).
- [3] Qu, S. & Sheng, P. Microwave and acoustic absorption metamaterials. *Phys. Rev. Appl.* **17**, 047001 (2022).
- [4] Ma, G., Yang, M., Xiao, S., Yang, Z. & Sheng, P. Acoustic metasurface with hybrid resonances. *Nat. Mater.* **13**, 873–878 (2014).
- [5] Romero-García, V. *et al.* Perfect and broadband acoustic absorption by critically coupled sub-wavelength resonators. *Sci. Rep.* **6**, 1–8 (2016).
- [6] Lissek, H., Boulandet, R. & Fleury, R. Electroacoustic absorbers: bridging the gap between shunt loudspeakers and active sound absorption. *J. Acoust. Soc. Am.* **129**, 2968–2978 (2011).
- [7] Ao, W., Ding, J., Fan, L. & Zhang, S.-y. A robust actively-tunable perfect sound absorber. *Appl. Phys. Lett.* **115**, 193506 (2019).
- [8] Ma, X. *et al.* Structural acoustic controlled active micro-perforated panel absorber for improving wide-band low frequency sound absorption. *Mech. Syst. Signal Process.* **178**, 109295 (2022).

- [9] Guo, X., Volery, M. & Lissek, H. Pid-like active impedance control for electroacoustic resonators to design tunable single-degree-of-freedom sound absorbers. *J. Sound Vib.* **525**, 116784 (2022).
- [10] Guo, X., Lissek, H. & Fleury, R. Improving sound absorption through nonlinear active electroacoustic resonators. *Phys. Rev. Appl.* **13**, 014018 (2020).
- [11] Huang, S. *et al.* Acoustic perfect absorbers via helmholtz resonators with embedded apertures. *J. Acoust. Soc. Am.* **145**, 254–262 (2019).
- [12] Donda, K. *et al.* Extreme low-frequency ultrathin acoustic absorbing metasurface. *Appl. Phys. Lett.* **115**, 173506 (2019).
- [13] Donda, K., Zhu, Y., Merkel, A., Wan, S. & Assouar, B. Deep learning approach for designing acoustic absorbing metasurfaces with high degrees of freedom. *Extreme Mech. Lett* **56**, 101879 (2022).
- [14] Huang, S. *et al.* Acoustic perfect absorbers via spiral metasurfaces with embedded apertures. *Appl. Phys. Lett.* **113**, 233501 (2018).
- [15] Li, Y. & Assouar, B. M. Acoustic metasurface-based perfect absorber with deep subwavelength thickness. *Appl. Phys. Lett.* **108**, 063502 (2016).
- [16] McKay, A., Davis, I., Killeen, J. & Bennett, G. J. Semsas: a compact super absorber optimised for broadband, low-frequency noise attenuation. *Sci. Rep.* **10**, 17967 (2020).
- [17] Boulvert, J., Gabard, G., Romero-García, V. & Groby, J.-P. Compact resonant systems for perfect and broadband sound absorption in wide waveguides in transmission problems. *Sci. Rep.* **12**, 1–13 (2022).
- [18] Sergeev, S., Lissek, H. & Fleury, R. Ultrabroadband sound control with deep-subwavelength plasmacoustic metalayers. *arXiv preprint arXiv:2209.13673* (2022).
- [19] Li, S. *et al.* Broadband perfect absorption of ultrathin conductive films with coherent illumination: Superabsorption of microwave radiation. *Phys. Rev. B* **91**, 220301 (2015).
- [20] Lee, T., Nomura, T. & Iizuka, H. Damped resonance for broadband acoustic absorption in one-port and two-port systems. *Sci. Rep.* **9**, 1–11 (2019).
- [21] Yang, M. & Sheng, P. Sound absorption structures: From porous media to acoustic metamaterials. *Annu. Rev. Mater. Res* **47**, 83–114 (2017).
- [22] Merkel, A., Theocharis, G., Richoux, O., Romero-García, V. & Pagneux, V. Control of acoustic absorption in one-dimensional scattering by resonant scatterers. *Appl. Phys. Lett.* **107**, 244102 (2015).
- [23] Yang, M. *et al.* Sound absorption by subwavelength membrane structures: A geometric perspective. *CR MECANIQUE* **343**, 635–644 (2015).
- [24] Mei, J. *et al.* Dark acoustic metamaterials as super absorbers for low-frequency sound. *Nat. Commun.* **3**, 1–7 (2012).
- [25] Jiménez, N. *et al.* Broadband quasi perfect absorption using chirped multi-layer porous materials. *AIP Adv.* **6**, 121605 (2016).

- [26] Jiménez, N., Romero-García, V., Pagneux, V. & Groby, J.-P. Rainbow-trapping absorbers: Broadband, perfect and asymmetric sound absorption by subwavelength panels for transmission problems. *Sci. Rep.* **7**, 1–12 (2017).
- [27] Long, H., Shao, C., Cheng, Y., Tao, J. & Liu, X. High absorption asymmetry enabled by a deep-subwavelength ventilated sound absorber. *Appl. Phys. Lett.* **118**, 263502 (2021).
- [28] Wu, X. *et al.* High-efficiency ventilated metamaterial absorber at low frequency. *Appl. Phys. Lett.* **112**, 103505 (2018).
- [29] Xiang, X., Tian, H., Huang, Y., Wu, X. & Wen, W. Manually tunable ventilated metamaterial absorbers. *Appl. Phys. Lett.* **118**, 053504 (2021).
- [30] Du, J. *et al.* Bilayer ventilated labyrinthine metasurfaces with high sound absorption and tunable bandwidth. *Sci. Rep.* **11**, 5829 (2021).
- [31] Tsuruta, R., Li, X., Yu, Z., Iizuka, H. & Lee, T. Reconfigurable acoustic absorber comprising flexible tubular resonators for broadband sound absorption. *Phys. Rev. Appl.* **18**, 014055 (2022).
- [32] Yang, M. *et al.* Subwavelength total acoustic absorption with degenerate resonators. *Appl. Phys. Lett.* **107**, 104104 (2015).

Reviewer #1 (Remarks to the Author):

The authors have addressed all of my comments from the initial review in a comprehensive manner. I am happy with the revised version. In my opinion, it can be accepted for publication.

Reviewer #2 (Remarks to the Author):

Reviewer #2 Attachment on the following page.

Generally, the authors have well addressed the items listed on previous comment. In the rebuttal file, the authors claimed that the manuscript only provides the proof-of-concept of a new kind of absorbing material while the practical application is not concerned. In this regard, whether the proposed absorber can achieve near-perfect absorption ($>99\%$) even in theoretical consideration, i.e. η approximates zero, which may be more constructive than current result of 84% (one cell). In this case, what is the temperature requirement of the cooling stack and the possible way of meeting such a requirement.

Further, in addition to the severe temperature condition, the practical application seems also to be hindered by the fabrication of the stack as pore radius of which is dozens of micron. 1) How the pore radius influences the absorption performance; 2) The fabrication details of the experimental sample are not mentioned in the manuscript. Some discussion on these issues should be provided.

Reviewer #3 (Remarks to the Author):

The reviewer appreciates the authors' meticulous and patient response to the comments. Most of my comments have been well addressed.

However, the reviewer still has one concern. As stated by my previous comment 1 "the thermoacoustic energy conversion is a well-known, extensively investigated physical phenomenon", as we all know that thermoacoustic effect can convert thermal energy into acoustic energy by creating sound, the temperature gradient can technically serve as a sound generator, the sound absorption would definitely be changed if a sound source is placed at the two ends of the structure to offset the existing sound. Since the use of sound source is a common active sound absorption method that has been realized in literature and achieved broadband ultra-low frequency sound absorption, have you compared these two similar methods? The authors mentioned that your method doesn't need sound sensors, but temperature sensors would be required as two thermal sources with specific temperature gradient is necessary in your system, could your system be regarded as another type of source-sensor-feedback system? That's why the reviewer thought the proposed structure is active, and the results should be compared with those achieved by active methods.

NCOMMS-22-53558

Frozen sound: An ultra-low frequency, ultra-broadband,
non-reciprocal acoustic absorber

Response letter

A. Maddi et al.

April 2023

Answers to Reviewers' comments

We thank the reviewers for their comments. In the following, the reviewers comments are displayed in black, the answers are provided just after in red. Regarding the text modification, black represents the original text and blue the proposed updates.

Reviewer 1

The authors have addressed all of my comments from the initial review in a comprehensive manner. I am happy with the revised version. In my opinion, it can be accepted for publication.

We are pleased to have answered all reviewer #1 inquiries, and with his positive assessment toward our manuscript.

Reviewer 2

Generally, the authors have well addressed the items listed on previous comment. In the rebuttal file, the authors claimed that the manuscript only provides the proof-of-concept of a new kind of absorbing material while the practical application is not concerned. In this regard, whether the proposed absorber can achieve near-perfect absorption (>99%) even in theoretical consideration, i.e. η approximates zero, which may be more constructive than current result of 84% (one cell). In this case, what is the temperature requirement of the cooling stack and the possible way of meeting such a requirement.

Further, in addition to the severe temperature condition, the practical application seems also to be hindered by the fabrication of the stack as pore radius of which is dozens of micron. 1) How the pore radius influences

the absorption performance; 2) The fabrication details of the experimental sample are not mentioned in the manuscript. Some discussion on these issues should be provided.

Comment 1 : the authors claimed that the manuscript only provides the proof-of-concept of a new kind of absorbing material while the practical application is not concerned. In this regard, whether the proposed absorber can achieve near-perfect absorption (>99%) even in theoretical consideration, i.e. η approximates zero, which may be more constructive than current result of 84% (one cell). In this case, what is the temperature requirement of the cooling stack and the possible way of meeting such a requirement.

Answer 1 : As provided in the main text, Eq.8 states that the absorption coefficients has a maximum value given by

$$\alpha_{max}^+ = \frac{1}{\eta^2 + 1}. \quad (1)$$

Consequently, the absorption clearly depends on η^2 , suggesting that a quasi-perfect absorption can be reached for not so small temperature ratio, for instance, a value of $\eta = 0.1$ is sufficient to reach an absorption of 99%. More generally, with a more sophisticated experimental set-up and especially with better heat exchangers, such that the cold temperature would reach the theoretical one (of liquid nitrogen), i.e $\eta = 0.25$, one can expect to have an absorption coefficients of 94%, which is indeed already high but can still be enhanced by different means:

- Finding a way to reach even lower temperature, e.g. by using liquid helium
- Setting the 'ambient' heat exchanger to a higher temperature (warmer left side), while still using liquid nitrogen on the right side, such that a smaller temperature ratio $\eta = \frac{T_{right}}{T_{left}}$ is reached. For instance, if the left side temperature is doubled (i.e $\approx 600K$), then $\eta = 0.125$ and $\alpha = 98\%$.
- Using two cells in series (or more), as it was performed experimentally here. This allows to have an overall smaller η , where $\eta_{2cells} = \eta_{1cell}^2$. Therefore, using better heat exchangers such that the cold temperature would approach that of liquid nitrogen, then $\eta_{2cells} = (0.25)^2$, and the absorption can theoretically reach 0.996%.

In our opinion, Methods 1 and 2 might be harder to put in practice compared to the last option, which is why we decided to investigate experimentally the last option in our study.

Modification 1 : Theoretical description Subsection, Lines 138

...

It is worth mentioning that a similar result can also be obtained by heating the left side of the stack rather than cooling its right side, but in order to reach the same absorption of about 0.94, a very high hot temperature of approximately 1200K (i.e $\eta = 0.25$) is needed, which is not easy to achieve in practice.

The aforementioned absorption can be further enhanced while still using liquid nitrogen: a potential solution is to use two cells in series, such that the velocity jump provided by the two cells is equal to the product of the two (i.e., $\eta_{2cells} = \eta_{1cell}^2$). Hence, an absorption of $\alpha^+ \approx 0.996$ can be achieved theoretically.

....

Comment 2 : How the pore radius influences the absorption performance

Answer 2 : The pore radius (r_s) is an important element since it must be smaller than the visco-thermal acoustic boundary layer thicknesses, and also because the viscous resistance depends on it :

The viscous resistance was derived in the Supplementary Information, Eq.6, and is given by,

$$R_v = \frac{8\rho_{m,0}\nu_{m,0}L_s}{\phi S r_s^2}$$

where, the dependence on r_s is clearly exhibited.

Additionally, in the main text, a normalized viscous resistance was introduced for ease of reading ($R = R_v/Z$), and it was stated that there exists an optimal value that maximize the absorption, and it was find to be

$$R_{opt} = 2\eta^2 - \eta + 1,$$

Therefore, an optimal value of pore radius exist. This value should be small enough to allow an isothermal contact and to set the normalized viscous resistance close to its optimum value.

Note that, the other parameters involved in the expression of R_v can also be used to tune the viscous resistance, like for instance the length of stack L_s or its porosity.

In the following, we propose a minor modification of the supplementary information :

Modification 2 : Supplementary Information I.B

.... Finally, combining both of the previous assumptions, namely $\omega L_s/c \ll 1$ and $r_s \ll \delta_{\kappa,\nu}$, leads after simplifications to the following transfer matrix of the stack :

$$\mathbf{M}_s \approx \begin{pmatrix} 1 & -\overbrace{\frac{8\rho_{m,0}\nu_{m,0}L_s}{\phi S r_s^2}}^{R_v} \\ 0 & \frac{T_C}{T_0} \end{pmatrix}. \quad (2)$$

where R_v represents the viscous resistance of the stack, which depends on both the stack geometry and the thermo-physical properties of the fluid. Furthermore, the resistivity parameter R used in the main article, is a normalized version of this viscous resistance, such that $R = R_v/Z$.

Comment 3 : The fabrication details of the experimental sample are not mentioned in the manuscript. Some discussion on these issues should be provided.

Answer 3 : The experimental sample used in this study consists of readily available industrial materials and were just reshaped to fit inside the waveguide. For instance, the meshgrids (bought from a company called Gantois) were simply cut by a scissor and several layers were stacked together to form the stack.

However, we do agree that no mention was given in the text, and we propose a the modification below:

Modification 3 : Methods section

The absorber consists of either one cell or two cells in series, depending on the chosen configuration. Each cell is equipped with a stack of length $L_s = 16$ mm, made of several stacked layers of stainless steel wire meshes (Manufactured by Gantois, toile metallique 304LR n°167 FR0 056), with an estimated porosity $\phi_s = 0.7$ and an estimated pore radius $r_s = 74$ μm . A cold and an ambient heat exchangers are attached to each side of the stack: they both consist of a honeycombed aluminum material with a length $L = 1.5$ cm,

Reviewer 3

The reviewer appreciates the authors' meticulous and patient response to the comments. Most of my comments have been well addressed. However, the reviewer still has one concern. As stated by my previous comment 1 "the thermoacoustic energy conversion is a well-known, extensively investigated physical phenomenon", as we all know that thermoacoustic effect can convert thermal energy into acoustic energy by creating sound, the temperature gradient can technically serve as a sound generator, the sound absorption would definitely be changed if a sound source is placed at the two ends of the structure to offset the existing sound. Since the use of sound source is a common active sound absorption method that has been realized in literature and achieved broadband ultra-low frequency sound absorption, have you compared these two similar methods? The authors mentioned that your method doesn't need sound sensors, but temperature sensors would be required as two thermal sources with specific temperature gradient is necessary in your system, could your system be regarded as another type of source-sensor-feedback system? That's why the reviewer thought the proposed structure is active, and the results should be compared with those achieved by active methods.

Comment 1 : As stated by my previous comment 1 "the thermoacoustic energy conversion is a well-known, extensively investigated physical phenomenon", as we all know that thermoacoustic effect can convert thermal energy into acoustic energy by creating sound, the temperature gradient can technically serve as a sound generator, the sound absorption would definitely be changed if a sound source is placed at the two ends of the structure to offset the existing sound. Since the use of sound source is a common active sound absorption method that has been realized in literature and achieved broadband ultra-low frequency sound absorption, have you compared these two similar methods?

Answer 1 :

Most of the active sound absorption (based on loudspeakers) have some limitations, mainly :

- Instabilities that may arise from the electroacoustic feedback.

- Resonant dependant (as a loudspeaker is essentially a mass-spring system).
- Low frequency : a limit on the displacement of the membrane.

The system proposed in our work allows to absorb very low frequencies (10Hz experimentally, and even lower if experimental facilities are available - see the discussion on the low frequency limit), with a larger bandwidth given its non-resonant nature, a feature that would be difficult to realize with an electroacoustic feedback.

However, to our knowledge, the only active method that has achieved a broadband and high absorption level is based on plasmacoustics. At the present time, this study is still undergoing peer review. Moreover, the authors of this study tackled a problem slightly different from ours, since their device is intended for a reflection problem (i.e., with a rigid backing), whereas here we address a problem where both reflection by and transmission through the two-port are possible.

Modification 1 : Discussion Section

To address this first comment of the referee, we propose to modify the manuscript as shown below with blue fonts. Also, note that in the introduction section, we have added some references of previous works dealing with sound absorption by loudspeakers equipped with feedback loop ([1, 2]) :

... In practice, such a velocity jump can be achieved by applying a strong cooling on one side of the porous material. The resulting two-port becomes non-reciprocal, which allows the control of unidirectional waves, but above all it allows to get a higher absorption which is only limited by the temperature ratio imposed along the meshgrids. **It is clear that the thermoacoustic effect offers the possibility to develop new kinds of absorbers capable of reaching high levels of absorption, even at extremely low frequencies. To the best of our knowledge, the only device able to deliver a similar pressure and velocity drop is an active system based on plasmacoustics, and has been recently used to design a broadband and highly effective absorber [3] in the reflection problem.**

The experiments were performed by using liquid nitrogen as the cold heat source and the results

Comment 2 : The authors mentioned that your method doesn't need sound sensors, but temperature sensors would be required as two thermal sources with specific temperature gradient is necessary in your system, could your system be regarded as another type of source-sensor-feedback system?

Answer 2 : The temperature sensors (Thermocouples) are used only to monitor the temperature, and to check that it's close to what we expect theoretically (for example, to check that the ambient side is close to the room temperature at the steady state). Therefore, no sort of feedback is used here (in other words, we could remove the temperature sensors without any impact on the operation of the system).

References

- [1] Koutserimpas, T. T., Rivet, E., Lissek, H. & Fleury, R. Active acoustic resonators with reconfigurable resonance frequency, absorption, and bandwidth. *Phys. Rev. Appl.* **12**, 054064 (2019).

- [2] Rivet, E., Karkar, S. & Lissek, H. Broadband low-frequency electroacoustic absorbers through hybrid sensor-/shunt-based impedance control. *IEEE Trans Control Syst Technol* **25**, 63–72 (2016).
- [3] Sergeev, S., Lissek, H. & Fleury, R. Ultrabroadband sound control with deep-subwavelength plasmacoustic metalayers. *arXiv preprint arXiv:2209.13673* (2022).

Reviewer #2 (Remarks to the Author):

This work has been well revised. I recommend its acceptance.

Reviewer #3 (Remarks to the Author):

The authors have addressed all of my comments. I'm happy with the revised version.

NCOMMS-22-53558

Frozen sound: An ultra-low frequency, ultra-broadband,
non-reciprocal acoustic absorber.

June 2, 2023

Answers to Reviewers' comments

Reviewer #2 (Remarks to the Author):

This work has been well revised. I recommend its acceptance.

We are pleased to have answered all reviewer #2 inquiries.

Reviewer #3 (Remarks to the Author):

The authors have addressed all of my comments. I'm happy with the revised version.

We are pleased to have answered all reviewer #3 inquiries.